# Timely initiation of breastfeeding and associated factors among mothers having children less than two years of age in sub-Saharan Africa: A multilevel analysis using recent Demographic and Health Surveys data

**Achamyeleh Birhanu Teshale** ⓘ *, **Getayeneh Antehunegn Tesema**

Department of Epidemiology and Biostatistics, Institute of Public Health, College of Medicine and Health Sciences, University of Gondar, Gondar, Ethiopia

* achambir08@gmail.com

## Abstract

### Background

Despite the significant advantages of timely initiation of breastfeeding (TIBF), many countries particularly low- and middle-income countries have failed to initiate breastfeeding on time for their newborns. Optimal breastfeeding is one of the key components of the SDG that may help to achieve reduction of under-five mortality to 25 deaths per 1000 live births.

### Objective

To assess the pooled prevalence and associated factors of timely initiation of breastfeeding among mothers having children less than two years of age in sub-Saharan Africa.

### Methods

We used pooled data from the 35 sub-Saharan Africa (SSA) Demographic and Health Surveys (DHS). We used a total weighted sample of 101,815 women who ever breastfeed and who had living children under 2 years of age. We conducted the multilevel logistic regression and variables with p<0.05, in the multivariable analysis, were declared significantly associated with TIBF.

### Results

The pooled prevalence of TIBF in SSA was 58.3% [95%CI; 58.0–58.6%] with huge variation between countries, ranging from 24% in Chad to 86% in Burundi. Both individual and community level variables were associated with TIBF. Among individual-level factors; being older-aged mothers, having primary education, being from wealthier households, exposure to mass media, being multiparous, intended pregnancy, delivery at a health facility, vaginal delivery, single birth, and average size of the child at birth were associated with higher odds

**Data Availability Statement:** All relevant data are within the paper and its Supporting information files.

**Funding:** the author(s) received no specific funding for this work.

**Competing interests:** The authors have declared that no competing interests exist.

**Abbreviations:** ANC, Antenatal Care; DHS, Demographic and Health Surveys; ICC, Intraclass Correlation Coefficient; MOR, Median Odds Ratio; PCV, Proportional Change in Variance; SSA, Sub Saharan Africa; TIBF, Timely Initiation of Breastfeeding; VIF, Variance inflation factor.

of TIBF. Of community-level factors, rural place of residence, higher community level of ANC utilization, and health facility delivery were associated with higher odds of TIBF.

## Conclusion

In this study, the prevalence of TIBF in SSA was low. Both individual and community-level factors were associated with TIBF. The authors recommend interventions at both individual and community levels to increase ANC utilization as well as health facility delivery that are crucial for advertising optimal breastfeeding practices such as TIBF.

## Background

Breastfeeding is one of the effective interventions that can reduce 55% to 87% of neonatal mortality and morbidity, particularly due to infections like diarrhea, neonatal sepsis, and pneumonia [1–5]. Globally, optimal breastfeeding can avoid the deaths of more than 800,000 under-fives annually. In lower and middle-income countries, an estimated 13% of all child deaths can be prevented if optimal breastfeeding is practiced [6]. Timely initiation of breastfeeding (TIBF) is giving breast milk to the newborn within one hour of birth [5]. This enables the newborn to take colostrum, which stimulates milk production and promote oxytocin release. In addition, taking colostrum helps the newborn to get protective factors such as antibodies [7]. TIBF can also facilitate bonding between the mother and her baby, reduce the incidence of postpartum hemorrhage, and ensure longer breastfeeding duration [8, 9]. Furthermore, TIBF reduces about 22% of neonatal deaths [1].

Despite the major public health implication of TIBF, many countries (especially low and middle-income countries) failed to initiate breastfeeding promptly for their newborns [10, 11]. Every year, in the world, about half of the newborns do not get breast milk in the first hour after delivery [12]. In sub-Saharan Africa (SSA), the prevalence of TIBF is 52.83% ranging from 17% in Guinea to 95% in Malawi [13, 14].

Studies conducted elsewhere revealed that factors such as maternal age, maternal education, wealth status, maternal occupation, place of birth, antenatal care (ANC) visit, mode of delivery, pregnancy intention, size of the child at birth, and place of residence are associated with TIBF [10, 15–21].

By 2030, the Sustainable Development Goal (SDG) aimed to reduce under-five mortality to 25 deaths per 1000 live births and one of the best strategy to achieve this plan is through increasing optimal feeding habits among children [22]. Besides, the 2010 Global Burden of Disease (GBD) identified suboptimal breastfeeding practice as the top three leading contributors of disease in Sub Saharan Africa [23]. The number of studies undertaken in sub-Saharan Africa did not involve the community-level factors related to TIBF. Therefore, this study aimed to assess the pooled prevalence and associated factors of timely initiation of breastfeeding in sub-Saharan Africa. The findings of this study could help policymakers to make a wise decision regarding optimal breastfeeding practices such as TIBF.

## Methods

### Data source

We used pooled data from the 35 SSA countries Demographic and Health Surveys (DHS), which were conducted from 2008–2019. All these surveys used a stratified two-stage cluster

sampling technique. The most recent DHS data was selected for analysis from each country specifically for those countries that have more than one surveys. For our study, we used kids data set with a total weighted sample of 101,815 women who ever breastfeed and who had living children under 2 years of age.

## Variables of the study

**Dependent variable.** The outcome variable was timely initiation of breastfeeding and it is giving breast milk to the newborn within one hour of birth. It was measured based on maternal report and coded as 1 "if the mother initiated breast milk within 1 hour" and 0 "otherwise".

**Independent variables.** Both individual and community level independent variables were incorporated in this study. The individual-level factors used in this study were; maternal age, maternal education, maternal occupation, marital status, household wealth status, mass media exposure, parity, pregnancy intention, ANC visit, place of delivery, mode of delivery, type of birth, size of the child at birth, and sex of the child. The Six community-level variables included were; place of residence, community-level media exposure, community level of women education, community poverty level, community level of ANC utilization, and community level of delivery at a health facility. The community-level factors (community level media exposure, community level of women education, community level of ANC utilization, community level of poverty, and community level of delivery at a health facility) were generated by aggregating individual-level factors, as these factors were not directly found from surveys (Table 1).

## Data management and statistical analysis

Appending data, extraction, re-coding, and statistical analysis were performed using Stata version 14 software. Sample weight was applied to adjust over or under sampling and we used the SVY command to account for the complex survey design and generalizability [24]. Because of the hierarchical nature of the DHS data, we conducted the multilevel analysis. While doing the multilevel analysis, we fitted four models: the null model (with only the outcome variable), Model I (containing individual-level factors only), model II (fitted with community-level factors only), and Model III (fitted with both individual and community-level factors). To examine clustering and the extent to which community-level factors explain the unexplained variance of the null model, the Intraclass correlation coefficient (ICC), a proportional change in variance (PCV), and median odds ratio (MOR) were checked. Model fitness was checked by deviance and the model with the lowest deviance was used as the best-fitted model. Variance inflation factor (VIF) was used to assess Multicollinearity and there was no Multicollinearity between independent variables, with a mean VIF of 1.81 (the minimum and the maximum VIF was 1.01 and 3.96 respectively). The bivariable analysis was used to select eligible variables for multivariable analysis (variables with a p-value <0.20 were eligible). Then, in the multivariable analysis, adjusted odds ratio (AOR) with 95% Confidence interval (CI) were reported, and variables with p<0.05 in the multivariable analysis were declared to be significantly associated with TIBF.

## Ethical consideration

Since this is a secondary DHS data analysis, ethical approval was not required. However, from the DHS online archive (www.dhsprogram.com), we requested the DHS datasets, obtained permission to access, and download the data files.

**Table 1. Categories/Description of independent variables.**

| Variables | Categorization and description of variables |
|---|---|
| **Individual-level variables** | |
| Maternal age | It is the current age of women categorized as 15–19, 20–14, 25–29, 30–34, 35–39, 40–44, and 45–49 in the DHS data set. |
| Maternal education | The level of education a woman achieved and categorized as no education, primary, secondary, and tertiary and above education |
| Maternal occupation | It is based on the working status of women and categorized in to working and not working |
| Marital status | The current marital status re-categorized as married and unmarried |
| Household wealth status | It is categorized as first (lowest), second, third (middle), fourth, and fifth (highest) wealth quantiles |
| Mass media exposure | Generated by combining whether a respondent reads the newspaper, listens to the radio, and watch television and coded as "yes" if the mother was exposed to at least one of the three media and "no" otherwise. |
| Parity | Re-categorized as Primiparous, multiparous, grand multiparous |
| Pregnancy intention | Re-categorized as intended (if the pregnancy was wanted) and unintended (incorporated both mistimed and unintended) |
| ANC visit | It is the number of ANC visits for the last pregnancy and re-categorized as no visit, 1 to 3 visit, and 4 & above visits |
| Place of delivery | The place where the mother gave the last birth and re-categorized as delivery at home and health facility |
| Mode of delivery | The route of delivery, which is categorized as delivery by cesarean section and delivery through vagina |
| Type of birth | Re-categorized as single and multiple (if the mother gave 2 or more child during the last birth) |
| Size of the child at birth | This is based on the maternal report about the size of the recent child and re-categorized as average, small, and large-sized baby |
| sex of the child | The sex of the last child and re-categorized as male and female |
| **Community-level variables** | |
| Place of residence | The area where the women live and categorized as rural residence and urban residence |
| Community-level of media exposure | A community-level variable measured by the proportion of women who had exposed to at least one media; television, radio, or newspaper and categorized based on national median value as low (communities with ≤ 50% of women exposed) and high (communities with >50% of women exposed) community-level media exposure. |
| Community-level of women education | Aggregate values measured by the proportion of women with a minimum of primary level of education derived from data on respondents' level of education. Then, it was categorized using national median value to values: low (communities with ≤ 50% of women have at least primary education) and high (communities with > 50% of women have at least primary education) community level of women education. |
| Community-level health facility delivery | Aggregate values measured by the proportion of women with health facility delivery and recoded as low (communities with ≤ 50% of women delivered at the health facility) and high (communities with >50% of women have delivered at health facility) community level of health facility delivery. |
| Community-level ANC utilization | Aggregate values measured by the proportion of women with a minimum of four or more ANC visits. We categorized it using national median value to values: low (communities with ≤ 50% of women have at four ANC visits) and high (communities with > 50% of women have at least four ANC visits) community level of ANC utilization. |
| Community poverty level | Aggregated variable from household wealth status (proportion of women from the first and second quantiles) and like the above community-level variables, it was recoded as low and high community poverty level. |

## Results

### Sociodemographic characteristics of respondents and newborns

The majority of the study participants were from Benin followed by the Democratic Republic of Congo (S1 Table). The median age of the respondents was 27 (IQR = 22–32) years. Most (59.54%) of the participants had some formal education and 48.19% of respondents were multiparous. Greater than half (53.64%) of respondents had four or more ANC visits and more than two-thirds (69.92%) of the respondents gave birth at the health facility. The majority (95.13%) of women gave birth through the vagina and 98.30% of the mother gave a single birth. Regarding place of residence, most (69.11%) of women were rural dwellers (Table 2).

### Prevalence of timely initiation of breastfeeding in sub-Saharan Africa

The pooled prevalence of timely initiation of breastfeeding in SSA was 58.3% [95%CI; 58.0–58.6%] with huge variation between countries, ranging from 24% [95%CI; 23–25%] in Chad to 86% [95%CI; 85–87%] in Burundi (Fig 1).

### Factors associated with timely initiation of breastfeeding in sub-Saharan Africa

**Fixed effect analysis.** We used the final model (Model III) to assess the factors associated with TIBF in SSA. All independent variables were eligible for multivariable analysis since all had a p-value <0.20. In the multivariable multilevel analysis, both individual and community-level variables were associated with TIBF. The odds of TIBF was higher among mothers whose age was 20–24, 25–29, 30–34, 35–39, 40–44, and 45–49 years as compared to those mothers whose age was young (15–19 years). The odds of TIBF was 1.25 [AOR = 1.25; 95%CI: 1.19–1.31] times higher among mothers who had primary education compared to those who had no formal education. Those mothers who were from households with third, fourth, and fifth wealth quantiles had 1.12 [AOR = 1.12; 95%CI: 1.05–1.20], 1.17 [AOR = 1.17; 95%CI: 1.08–1.25], and 1.36 [AOR = 1.36; 95%CI: 1.24–1.49] times higher odds of TIBF as compared to those from households with first wealth quantile. Mothers who have been exposed to mass media had 12% [AOR = 0.88; 95%CI: 0.83–0.92] lower odds of TIBF as compared to their counterparts. The odds of TIBF was 1.15 [AOR = 1.15; 95%CI: 1.09–1.22] times higher among multiparous women as compared to Primiparous women. Regarding pregnancy intention, mothers whose pregnancy was unintended had 7% [AOR = 0.93; 95%CI: 0.89–0.98] lower odds of TIBF as compared to those whose pregnancy was intended. Looking at the place and mode of delivery, mothers who were delivered at the health facility and those who were delivered through cesarean section had 1.73 [AOR = 1.73; 95%CI: 1.64–1.83] times higher and 72% [AOR = 0.28; 95%CI: 0.25–0.31] lower odds of TIBF respectively as compared to their counterparts. The odds of TIBF was 27% [AOR = 0.73; 95%CI: 0.63–0.83] lower among mothers who gave multiple births as compared to those who gave a single birth. Mothers who gave small and large-sized babies had 23% [AOR = 0.77; 95%CI: 0.73–0.81] and 17% [AOR = 0.83; 95%CI: 0.79–0.87] lower odds of TIBF respectively as compared to those who gave the average-sized baby. Among community-level factors, mothers from the rural area had 1.43 [AOR = 1.43; 95%CI: 1.33–1.53] times higher odds of TIBF as compared with those from urban areas. Mothers from communities with higher community levels of ANC utilization and health facility delivery had 1.08 [AOR = 1.08; 95%CI: 1.02–1.14] and 1.10 [AOR = 1.10; 95%CI: 1.03–1.17] times higher odds of TIBF respectively as compared with their counterparts (Table 3).

**Table 2. Sociodemographic characteristics of respondents and newborns.**

| Variables | Frequency (N = 101,815) | Percentage (%) |
|---|---|---|
| **Individual-level variables** | | |
| Maternal age (years) | | |
| 15–19 | 10705 | 10.51 |
| 20–24 | 25489 | 25.03 |
| 25–29 | 27384 | 26.90 |
| 30–34 | 19628 | 19.28 |
| 35–39 | 12638 | 12.41 |
| 40–44 | 4871 | 4.78 |
| 45–49 | 1100 | 1.08 |
| Maternal educational | | |
| No education | 41190 | 40.46 |
| Primary | 33692 | 33.09 |
| Secondary | 24125 | 23.69 |
| Tertiary and above | 2808 | 2.76 |
| Marital status | | |
| Married | 89572 | 87.97 |
| Unmarried | 12243 | 12.03 |
| Maternal occupation | | |
| Working | 70423 | 69.17 |
| Not working | 31392 | 30.83 |
| Wealth quantiles | | |
| First | 22984 | 22.57 |
| Second | 22138 | 21.74 |
| Third | 20699 | 20.33 |
| Fourth | 19476 | 19.13 |
| Fifth | 16518 | 16.22 |
| Mass media exposure | | |
| No | 37163 | 36.50 |
| Yes | 64652 | 63.50 |
| Parity | | |
| Primiparous | 21891 | 21.50 |
| Multiparous | 49061 | 48.19 |
| Grand multiparous | 30863 | 30.31 |
| Pregnancy intention | | |
| Intended | 73785 | 72.47 |
| Unintended | 28030 | 27.53 |
| ANC visits | | |
| No visit | 9979 | 9.80 |
| 1 to 3 visit | 37219 | 36.56 |
| 4 & above visits | 54617 | 53.64 |
| Place of delivery | | |
| Home | 30628 | 30.08 |
| Health facility | 71187 | 69.92 |
| Mode of delivery | | |
| Cesarean section | 4957 | 4.87 |
| Vaginal | 96858 | 95.13 |
| Type of birth | | |
| Single | 100083 | 98.30 |
| Multiple | 1732 | 1.70 |

*(Continued)*

**Table 2.** (Continued)

| Variables | Frequency (N = 101,815) | Percentage (%) |
|---|---|---|
| Size of the child at birth | | |
| Average | 49773 | 48.89 |
| Small | 16316 | 16.03 |
| Large | 35726 | 35.09 |
| Sex of child | | |
| Male | 51292 | 50.38 |
| Female | 50522 | 49.62 |
| **Community-level variables** | | |
| Residence | | |
| Urban | 31448 | 30.89 |
| Rural | 70367 | 69.11 |
| Community-level of women education | | |
| Low | 50465 | 49.57 |
| High | 51350 | 50.43 |
| Community poverty level | | |
| Low | 51209 | 50.30 |
| High | 50606 | 49.70 |
| Community-level ANC utilization | | |
| Low | 50179 | 49.28 |
| High | 51636 | 50.72 |
| Community-level delivery at a health facility | | |
| Low | 50670 | 49.77 |
| High | 51145 | 50.23 |
| Community-level of media exposure | | |
| Low | 50885 | 49.98 |
| High | 50930 | 50.02 |

**Random effect analysis.** Table 4 revealed the random effect analysis. The ICC and the MOR in the null model support the presence of significant variations of TIBF between clusters and countries. For example, the higher MOR value (1.37) in the null model indicates that if we randomly choose women from two different clusters, a woman from a cluster with higher rates of TIBF had 1.37 times higher odds of TIBF as compared to a woman who came from a cluster with lower rates of TIBF. Furthermore, the higher PCV in the final model revealed most of the variations of TIBF were attributable to both individual and community-level factors. Moreover, as shown in Table 4, the best-fitted model was the final model (model III) since it had the lowest deviance (Table 4).

## Discussion

This study aimed to assess timely initiation of breastfeeding and associated factors in SSA using multilevel analysis. The pooled prevalence of TIBF in SSA was 58.3%. When we compared with different individual studies, this figure is in line with studies conducted in Ethiopia and Western Nepal [15, 25]. The finding is higher than the findings from SSA, Pakistan, India, and Bangladesh [14, 26–28]. The prevalence found in this study is lower than studies conducted in Ethiopia [29] and Nepal [30]. The divergence of this finding from other studies may be attributable to discrepancies in access to health facilities between countries. The other

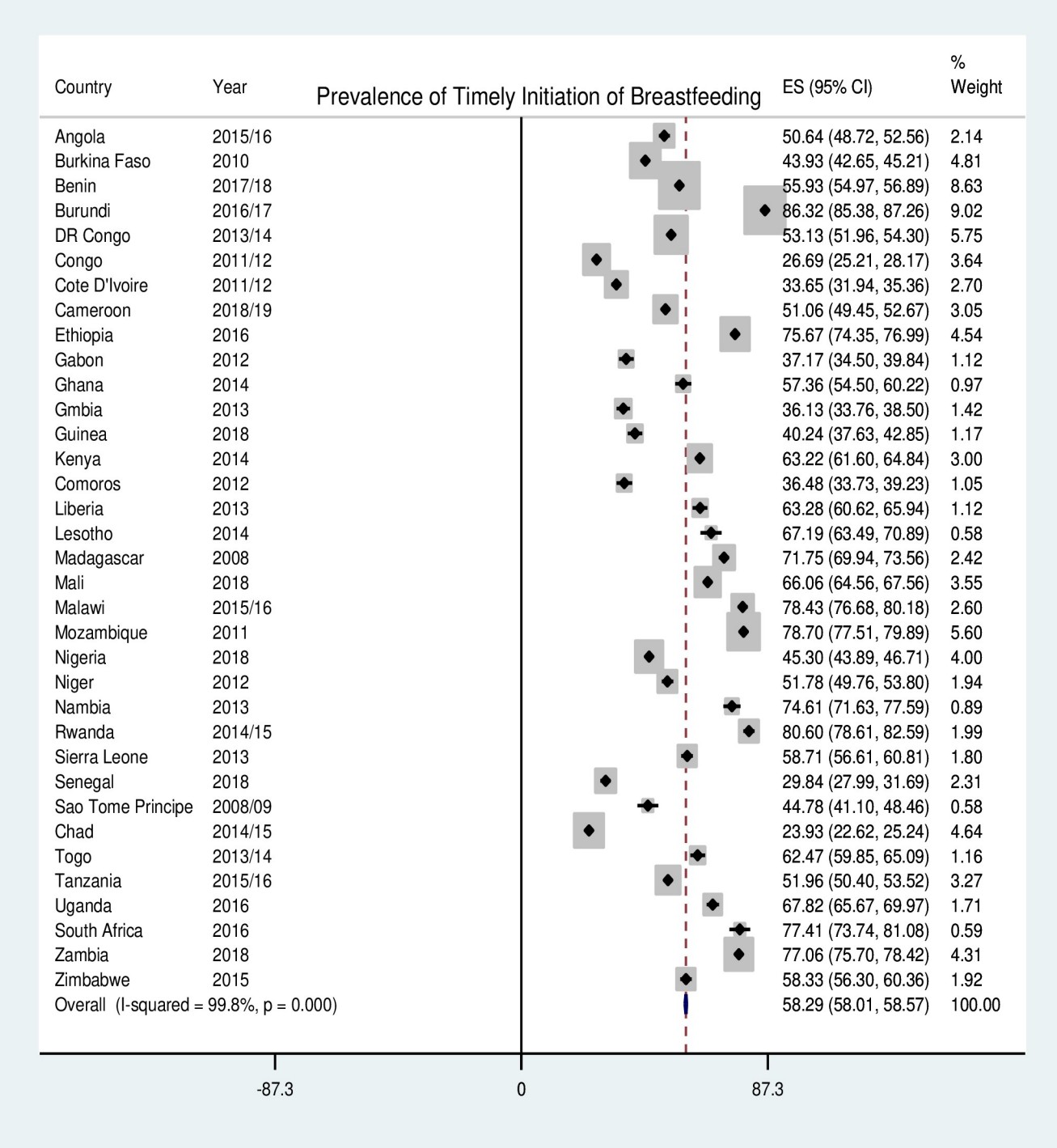

**Fig 1. Forest plot showing the pooled prevalence of TIBF in SSA countries.**

possible reason may be due to the variations in sociodemographic features, and socio-cultural practices between countries. The difference in sample size (since most of the studies were based on a single country) and study period might be the other possible explanation of the discrepancy between our findings and other studies' findings.

**Table 3. Multivariable multilevel analysis for factors associated with TIBF in SSA.**

| Variables | Null model | Model I AOR[95%CI] | Model II AOR[95%CI] | Model III AOR[95%CI] |
|---|---|---|---|---|
| Maternal age (years) | | | | |
| 15–19 | | 1.00 | | 1.00 |
| 20–24 | | 1.09[1.02–1.17] | | 1.09[1.01–1.17] * |
| 25–29 | | 1.21[1.11–1.31] | | 1.21[1.11–1.31] *** |
| 30–34 | | 1.26[1.15–1.38] | | 1.26[1.15–1.38] *** |
| 35–39 | | 1.30[1.17–1.45] | | 1.31[1.17–1.45] *** |
| 40–44 | | 1.31[1.16–1.49] | | 1.32[1.16–1.49] *** |
| 45–49 | | 1.29[1.04–1.58] | | 1.28[1.04–1.58] * |
| Maternal educational | | | | |
| No education | | 1.00 | | 1.00 |
| Primary | | 1.24[1.18–1.30] | | 1.25[1.19–1.31] *** |
| Secondary | | 0.95[0.89–1.01] | | 0.99[0.94–1.06] |
| Tertiary and above | | 0.95[0.83–1.08] | | 1.01[0.87–1.14] |
| Marital status | | | | |
| Married | | 0.92[0.87–0.98] | | 0.90[0.82–1.01] |
| Unmarried | | 1.00 | | 1.00 |
| Maternal occupation | | | | |
| Working | | 0.87[0.83–0.91] | | 0.86[0.77–1.02] |
| Not working | | 1.00 | | 1.00 |
| Household wealth quantiles | | | | |
| First | | 1.00 | | 1.00 |
| Second | | 1.02[0.96–1.09] | | 1.03[0.97–1.09] |
| Third | | 1.08[1.02–1.15] | | 1.12[1.05–1.20] *** |
| Fourth | | 1.04[0.97–1.12] | | 1.17[1.08–1.25] *** |
| Fifth | | 1.10[1.01–1.19] | | 1.36[1.24–1.49] *** |
| Mass media exposure | | | | |
| No | | 1.00 | | 1.00 |
| Yes | | 0.86[0.82–0.91] | | 0.88[0.83–0.92] *** |
| Parity | | | | |
| Primiparous | | 1.00 | | 1.00 |
| Multiparous | | 1.13[1.07–1.20] | | 1.15[1.09–1.22] *** |
| Grand multiparous | | 1.03[0.95–1.12] | | 1.03[0.95–1.12] |
| Pregnancy intention | | | | |
| Intended | | 1.00 | | 1.00 |
| Unintended | | 0.93[0.89–0.97] | | 0.93[0.89–0.98] ** |
| ANC visits | | | | |
| No visit | | 1.00 | | |
| 1 to 3 visit | | 1.01[0.93–1.09] | | 0.99[0.92–1.09] |
| 4 & above visits | | 1.01[0.93–1.09] | | 1.01[0.93–1.09] |
| Place of delivery | | | | |
| Home | | 1.00 | | 1.00 |
| Health facility | | 1.68[1.60–1.77] | | 1.73[1.64–1.83] *** |
| Mode of delivery | | | | |
| Cesarean section | | 0.28[0.25–0.30] | | 0.28[0.25–0.31] *** |
| Vaginal | | 1.00 | | 1.00 |
| Type of birth | | | | |
| Single | | 1.00 | | 1.00 |

(*Continued*)

**Table 3.** (Continued)

| Variables | Null model | Model I AOR[95%CI] | Model II AOR[95%CI] | Model III AOR[95%CI] |
|---|---|---|---|---|
| Multiple | | 0.72[0.63–0.83] | | 0.73[0.63–0.83] *** |
| Size of the child at birth | | | | |
| Average | | 1.00 | | 1.00 |
| Small | | 0.77[0.73–0.81] | | 0.77[0.73–0.81] *** |
| Large | | 0.82[0.79–0.86] | | 0.83[0.79–0.87] *** |
| Sex of child | | | | |
| Male | | 1.00 | | 1.00 |
| Female | | 1.03[0.99–1.06] | | 1.02[0.99–1.06] |
| Residence | | | | |
| Urban | | | 1.00 | 1.00 |
| Rural | | | 1.21[1.14–1.28] | 1.43[1.33–1.53] *** |
| Community-level of women education | | | | |
| Low | | | 1.00 | 1.00 |
| High | | | 1.01[0.96–1.07] | 1.01[0.95–1.06] |
| Community poverty level | | | | |
| Low | | | 1.00 | 1.00 |
| High | | | 0.99[0.94–1.05] | 1.01[0.95–1.07] |
| Community-level ANC utilization | | | | |
| Low | | | 1.00 | 1.00 |
| High | | | 1.08[1.02–1.14] | 1.08[1.02–1.14] * |
| Community-level delivery at the health facility | | | | |
| Low | | | 1.00 | 1.00 |
| High | | | 1.01[0.95–1.07] | 1.10[1.03–1.17] ** |
| Community-level of media exposure | | | | |
| Low | | | 1.00 | 1.00 |
| High | | | 1.01[0.94–1.05] | 1.02[0.97–1.09] |

Note:

*** = p<0.001,

** = p≤0.01,

* = pvalue<0.05.

In the multilevel multivariable analysis, both individual and community-level factors were associated with TIBF. Being in the older age group had higher odds of TIBF as compared with the younger age group. This is in concordance with studies done in Tanzania [17]. This is because older mothers might have experience in everything during previous pregnancies and

**Table 4. Random effect analysis and model fitness for assessing factors associated with TIBF in SSA.**

| Parameters | Null model | Model I | Model II | Model III |
|---|---|---|---|---|
| Community level variance[SE] | 0.111[0.010] | 0.109[0.010] | 0.103[0.009] | 0.102[0.009] |
| ICC | 0.033 | 0.032 | 0.030 | 0.030 |
| MOR | 1.37[1.34–142] | 1.36[1.33–1.41] | 1.35[1.32–1.39] | 1.35[1.32–1.39] |
| PCV | Reference | 0.02 | 0.07 | 0.08 |
| Model fitness | | | | |
| Log-likelihood | -69112.78 | -67283.28 | -69023.89 | -67087.79 |
| Deviance | 138225.56 | 134566.56 | 138047.78 | 134175.58 |

childbirths and more likely to be exposed to information regarding optimal breastfeeding practices [31]. Mothers who had formal education were more likely to start breastfeeding timely for their newborns. This is consistent with studies done in Nigeria and Tanzania [16, 17, 30]. This may be because education plays an important role in shifting mothers' views and behaviors about breastfeeding, maximizing ANC follow-up and raising the probability of delivery at health institutions [32, 33]. The impact of education may also be explained by the possibility that educated mothers would readily receive and comprehend health promotion messages such as infant feeding styles [34, 35].

Consistent with studies conducted elsewhere [18–20], mothers from third, fourth, and fifth household wealth quantiles had higher odds of TIBF as compared with mothers from the first household wealth quantile. This might be because mothers from wealthy households have easy access to education and maternal health care services such as institutional delivery services that enforce the practice of TIBF [36, 37].

Mothers who have been exposed to mass media had lower odds of TIBF as compared to their counterparts. This might be due to the aggressive advertising of infant formula feedings, milk substitutes, teats, and bottles in different media recently [38, 39]. This may also mean that these media (radio, television, and newspaper) were not readily available and sufficient to encourage appropriate breastfeeding practices. However, the authors suggest further inquiry in this regard.

The study at hand also revealed that multiparous women were more likely to start breastfeeding timely as compared to Primiparous mothers. This is in agreement with the findings of different studies conducted in Saudi Arabia, Nigeria, Ethiopia, and low and middle-income countries [15, 19, 40–42]. This could be because if a mother has more birth experience, it is more probable that the next baby would be put in the breast within 1 hour of birth, as through her successive pregnancies and deliveries the mother will be exposed to information on appropriate breastfeeding practices [43].

Delivery at the health facility was associated with higher odds of TIBF, in this study. This is in concordance with studies done in Ethiopia [21, 29], Tanzania [44], and Nepal [30]. This is an expected finding since many of the Health Care Centers and Hospitals already have certified midwives or any other qualified professionals available to enable and assist the mother to start breastfeeding early during childbirth.

Mode of delivery was another factor that was associated with TIBF in this study. Mothers who gave birth by cesarean section had lower odds of TIBF as compared to those who gave birth vaginally. This is in line with studies done in Ethiopia [21, 45], Nigeria [19], Turkey [46], Saudi Arabia [47], Lebanon [48], Brazil [49], and India [28]. The possible explanation is both the newborn delivered by cesarean section and the mothers who gave birth by cesarean section typically remain under various obstetric-related health conditions such as general anesthesia effect, pain, and fatigue [50]. This finding can also be attributed to long postoperative care, which delays mother-baby contact [51]. This result suggests that midwives should be aware of the negative relationship between cesarean delivery and breastfeeding initiation to mitigate delayed breastfeeding initiation in mothers with cesarean delivery.

Mothers with unintended pregnancies were less likely to initiate breastfeeding within 1 hour. This is in agreement with a study done in Ethiopia [52]. The possible reason may be mothers with unintended pregnancies are less likely to utilize maternal health services and due to this, they might not gain information regarding appropriate breastfeeding practices [53]. Moreover, women who experienced mistimed pregnancy might lose support from their families or partners for good healthcare-seeking behaviors of their children [54].

In this study, the type of birth was significantly associated with TIBF. Mothers with multiple births were less likely to initiate breastfeeding within 1 hour. This finding is supported by a

study conducted in low and middle-income countries [41]. The possible explanation is mothers with multiple pregnancy are more likely to give birth by a cesarean section [55], which in turn increase delayed initiation of breastfeeding.

Congruent with studies conducted in low and middle-income countries [41], Namibia [56], and Zimbabwe [42], being mothers with small and large-sized babies at birth was associated with lower odds of TIBF as compared to mothers who gave an average-sized baby. The reason behind this might be, in most cases, babies with abnormal weights might be separated from their mothers for longer periods after delivery as they may suffer from other comorbid conditions that need intervention [57]. This separation results in the babies not being able to access breast milk early.

Moreover, community-level factors were associated with TIBF. Mothers from rural areas had higher odds of TIBF. This is a surprising finding that is in line with a study done in Zimbabwe [42] and Malawi [58]. The plausible explanation is the extended health extension program in remote areas to encourage women of reproductive age to utilize maternal health services [59], which in turn helps them to gain information regarding optimal breastfeeding practices. However, it is an unusual finding and we recommend a further investigation in this regard.

Antenatal care and institutional delivery are the best opportunity to promote and educate mothers on essential healthy behaviors like appropriate newborn feeding practices [60]. The study at hand also revealed that, the higher number of women who had ANC visits and who gave birth at the health facility in a community, the more likely to develop a norm that encourages appropriate breastfeeding practices for the newborns such as TIBF. This suggests that improving community participation to increase maternal and child health service utilization can improve optimal breastfeeding practices, such as early breastfeeding initiation [21, 61].

The current study was conducted in SSA by using a multilevel analytical approach, which identifies factors associated with TIBF at both individual and community levels. Moreover, the results are representative of the entire SSA countries because we used appropriate analysis techniques such as weighting and multilevel analysis to get appropriate statistical estimates. Therefore, this study could help policymakers and responsible bodies to plan appropriate strategies and implement interventions. Despite the use of the nationally-representative data of each country, the study was not without limitation. Since the outcome was assessed based on the maternal report, there is a possibility of recall bias. Besides, the DHS did not collect some information such as maternal beliefs and knowledge towards breastfeeding so there may be residual confounding. In addition, we are unable to do a three level analysis, to account the country level heterogeneity, due to the convergence problem. Moreover, since it was a cross-sectional study, we are unable to show the cause and effect relationship between TIBF and independent variables.

## Conclusion

The prevalence of TIBF in SSA was low, considering the huge variation between countries and according to the WHO recommendation that all babies should benefit from breastfeeding early in life. Both individual and community level variables were associated with TIBF. Therefore, emphasis should be given to young women, women with poor socioeconomic status, mothers with lower parity, mothers who delivered by cesarean section, women who gave birth at home, and mothers who gave multiple births to plan appropriate strategies and implement interventions.

## Supporting information

**S1 Table. The 35 SSA countries used for analysis and their sample size.**
(DOCX)

## Acknowledgments

We would like to acknowledge the MEASURE DHS program, which helps us to access and use the data sets.

## Author Contributions

**Conceptualization:** Achamyeleh Birhanu Teshale, Getayeneh Antehunegn Tesema.

**Data curation:** Achamyeleh Birhanu Teshale, Getayeneh Antehunegn Tesema.

**Formal analysis:** Achamyeleh Birhanu Teshale, Getayeneh Antehunegn Tesema.

**Investigation:** Achamyeleh Birhanu Teshale, Getayeneh Antehunegn Tesema.

**Methodology:** Achamyeleh Birhanu Teshale, Getayeneh Antehunegn Tesema.

**Resources:** Achamyeleh Birhanu Teshale, Getayeneh Antehunegn Tesema.

**Software:** Achamyeleh Birhanu Teshale, Getayeneh Antehunegn Tesema.

**Validation:** Achamyeleh Birhanu Teshale, Getayeneh Antehunegn Tesema.

**Visualization:** Achamyeleh Birhanu Teshale, Getayeneh Antehunegn Tesema.

**Writing – original draft:** Achamyeleh Birhanu Teshale, Getayeneh Antehunegn Tesema.

**Writing – review & editing:** Achamyeleh Birhanu Teshale, Getayeneh Antehunegn Tesema.

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
