## [Decision Letter · Decision Letter 0]

25 Nov 2020

PONE-D-20-27660

Timely initiation breastfeeding and its associated factors among mothers with under 24 months living children in Sub Saharan Africa: a multilevel analysis using DHS Data from 35 Sub Saharan African countries

PLOS ONE

Dear Dr. Teshale,

Thank you for submitting your manuscript to PLOS ONE. After careful consideration, we feel that it has merit but does not fully meet PLOS ONE’s publication criteria as it currently stands. Therefore, we invite you to submit a revised version of the manuscript that addresses the points raised during the review process. Proofreading of the written English is deemed necessary throughout the manuscript.

We look forward to receiving your revised manuscript.

Kind regards,

Bárbara Hatzlhoffer Lourenço, Ph.D.

Academic Editor

PLOS ONE

Journal Requirements:

Reviewers' comments:

Reviewer's Responses to Questions

**Comments to the Author**

1. Is the manuscript technically sound, and do the data support the conclusions?

Reviewer #1: Yes

Reviewer #2: Yes

2. Has the statistical analysis been performed appropriately and rigorously? 

Reviewer #1: I Don't Know

Reviewer #2: Yes

3. Have the authors made all data underlying the findings in their manuscript fully available?

Reviewer #1: Yes

Reviewer #2: Yes

4. Is the manuscript presented in an intelligible fashion and written in standard English?

Reviewer #1: No

Reviewer #2: Yes

5. Review Comments to the Author

Reviewer #1: The manuscript submitted by Achamyeleh Birhanu Teshale and Getayeneh Antehunegn Tesema aims to investigate timely initiation breastfeeding and its associated factors in Sub Saharan African countries. This manuscript presents results that would be of interest to the community of scientists and clinicians concerned with delay initiation breastfeeding rates. Prior to publication, the following points should be addressed.

Title:

The title could be more succinct. Note that the location (Sub Saharan Africa) is repeated.

Abstract:

-Please, note that the abstract exceeds the number of words allowed.

-Kindly check if all the factors you described in the results section are community level factors. See that you presented “Among community level factors” and “of community level factors”.

-The conclusion paragraphs repeat the information presented in results.

Background:

-Please, consider if in addition to the ranges it is possible to inform the prevalence of TIBF in SSA.

-Does “Residence” mean area of residence?

-When you explain that previous studies only considered a single country, are you talking about studies performed in Africa? Please review this paragraph. I do not think that the limitation of the previous studies has a connection with the previous sentence.

Methods:

-What exactly does “KR” mean? Kids, what else?

-In the dependent variable paragraph, please, add the definition of TIBF and how it was measured. This information was mentioned in the discussion section (maternal report).

-In the independent variables paragraph, could you please add the categorization used for each variable in the analyses?

-Why did you categorize maternal age in that way? Please, explain the rationale for those ranges. Consider recategorizing the variable into less categories since in the discussion section you describe just "older and younger mothers".

-Please, note that in the independent variables paragraph you said “four community level variables” but there were 5 mentioned (including residence).

-Regarding the community level factors, based on what did you set a 50% cut-off point?

-Please, note that in table 1 you also presented a “Community poverty level” that was not described in methods section.

-In table 3, What does AOR mean? It was not mentioned in statistical analysis section. Besides, is the confidence interval 95%? Please add this information.

Results:

-The following information was given before: “total weighted sample of 101815 women was used for our analysis.”

-Please check the following sentence “Most (40.46%) of the respondents had no formal education” note that most of the participants (59.5% ) had some formal education.

-Observe that the main purpose of the study is to investigate associated factor of TIBF. However, in this section you used the following title: “Determinants of timely initiation of breastfeeding…”

-Concerning “…those from a higher risk cluster had 1.37 times higher odds of TIBF as compared to those individuals who come from the lower risk cluster” Please, clarify what a higher risk refers to? In this case higher odds of TIBF would be beneficial/expected, right? It seems contradictory to talk about higher risk of having 1.37 times higher odds of TIBF. See if you can reformulate the sentence.

-The information given from line 172 to 177 is repeating what was described from line 178 on.

-In table 3, I think that null model column is not necessary.

-Please, review the labels of the categories in the wealth index. You can name the quintiles from the first (lowest) to the fifth (highest), instead of “the lowest and the second quintile”. It will be useful for describing this factor in the results and discussion section.

Discussion and conclusion:

-Please, conclude the first paragraph of the discussion. You compared the results of the study with other literature, but what does 58% of TIBF mean? Does it comply with the recommendation? (WHO reference: Infant and Young Child Feeding. A tool for assessing national practices, policies and programmes. Geneva. 2003).

-The text from line 214 to 219 has already been described in the results section.

-In relation to “This is because educated mothers might have exposure to appropriate breastfeeding practices” could you explain it better?

In relation to “Contrary to our expectation, mothers who have been exposed to media had lower odds of TIBF as compared to their counterparts.” Why “contrary to our expectation”? What about the aggressive advertising of infant formula industry, milk substitutes, teats, and bottles regarding the media?

-The discussion on community levels of ANC utilization and health facility delivery is not clear. Can you give more details and expand the discussion?

-Line 292, determinants of TIBF, or associated factors?

-It is important that the discussion includes not only the comparison of the results with other studies and the authors conclusions, but also the theoretical basis for these conclusions. Please, support all your conclusions with bibliographic references.

-Please, review the conclusion: what was found in the study and the recommended "special emphasis", repeats the information.

References and general comments:

-Please, check all the references and make sure the institutions names are consistently cited. (Organization WH., WHO., World Health Organization, Unicef., UNICEF.?)

-Please, remove the space between lines (for example, between line 56 and 57). Check all the text and make sure the format is consistent.

-Please, avoid the following text structure (mistimed/unintended, inappropriate/suboptimal, physicians/midwives). If possible, choose one word or use “and” when necessary.

-Overall: I recommend editing the text to achieve a more appropriate grammar structure and scientific language.

Reviewer #2: Thanks for the opportunity to review this interesting paper. The paper is well-written the methodology is very clear and the results are well presented. My suggestions are as follows:

Abstract

Line 19 …’despite its major implication…’ perhaps it should read ‘despite THE major implications of…’ – However, I would suggest having this first line revised in its totality.

Methodology

The authors have stated that they selected DHS data from 35 countries. Malawi, for example, has the most recent DHS data in 2015-16, why did the authors choose to analyse the 2011 (which I suppose should be 2010 data) instead of the most recent data?

The analysis is a pooled one, with 35 different countries having their respective weighting variables according to their sample selection. It would be appropriate for the authors to describe whether they generated pooled weights and briefly explain how this was done.

Did the authors check for multicollinearity of their models?

Under the data analysis section, authors aught to mention that adjusted odds ratio and 95% CI were reported…

Results

This is just a 'cosmetic' comment in the presentation of the results: It would look great if the fixed effects are presented first then random effects later thus Table 2 could be Table 3 while the current Table 3 may be changed to Table 2.

Discussion

Line 285-290 – More explanation needed and if possible, provide some refs. Why would women coming from communities with high ANC utilization more likely to develop a norm of early BF initiation?

In the conclusion section, the authors have mention that the prevalence of TIBF was relatively lower, ,, lower compared to what? What are they comparing this prevalence to?

6. PLOS authors have the option to publish the peer review history of their article (what does this mean?). If published, this will include your full peer review and any attached files.

Reviewer #1: No

Reviewer #2: No

---

## [Author Response · Author response to Decision Letter 0]

8 Dec 2020

Date: December 2020

Point by point response to editor and reviewers comment

Title: Timely initiation breastfeeding and its associated factors among mothers with under 24 months living children in Sub Saharan Africa: a multilevel analysis using DHS Data from 35 Sub Saharan African countries

Manuscript number: PONE-D-20-27660

Dear editor and Reviewers: We really thank you for your valuable comments for the betterment of our manuscript. Your concerns and questions as well as suggestions are addressed in the revised manuscript (see the point-by-point response). 

Response to editor comment

Author’s response: Thank you. The revised manuscript is prepared according to PLOS ONE's style.

2. We suggest you thoroughly copyedit your manuscript for language usage, spelling, and grammar.

Author’s response: The revised manuscript is extensively edited for language usage, spelling, and grammar after consulting our colleagues who had MA degree in “teaching English as foreign language” and who had many years experience in the area of literature at University of Gondar. A copy of our manuscript showing the changes is indicated by using track changes (See supporting information file). 

Response to reviewers comment 

Reviewer #1:

1. Title:

The title could be more succinct. Note that the location (Sub Saharan Africa) is repeated.

Author’s response: Thank you for the important comment you raised. We amended the title accordingly in the revised manuscript. 

2. Abstract:

-Please, note that the abstract exceeds the number of words allowed.

Author’s response: Thank you. According to PLOS ONE journal submission guideline, the number of words in the abstract should not be above 500 and we prepared based on this and the total words in the abstract is below 500 (around 360) in the revised manuscript.

-Kindly check if all the factors you described in the results section are community level factors. See that you presented “Among community level factors” and “of community level factors”.

Author’s response: Thank you. It was to mean “Among individual level factors” and we amended it in the revised manuscript.

-The conclusion paragraphs repeat the information presented in results.

Author’s response: Thank you. We consider your comment in the revised manuscript. 

3. Background:

-Please, consider if in addition to the ranges it is possible to inform the prevalence of TIBF in SSA.

Author’s response: Thank you. We incorporated the prevalence of TIBF in SSA in the revised manuscript (see line 61 & 62). 

-Does “Residence” mean area of residence?

Author’s response: Yes, it was to mean “place of residence” or “area of residence” and we amend residence to place of residence in the revised manuscript.

-When you explain that previous studies only considered a single country, are you talking about studies performed in Africa? Please review this paragraph. I do not think that the limitation of the previous studies has a connection with the previous sentence.

Author’s response: Thank you. We amended these statements in the revised manuscript (see line 67-72). 

4. Methods:

-What exactly does “KR” mean? Kids, what else?

Author’s response: Thank you. It was to mean Kids data set and in the revised manuscript, we remove the abbreviation since we hope it add nothing. 

-In the dependent variable paragraph, please, add the definition of TIBF and how it was measured. This information was mentioned in the discussion section (maternal report).

Author’s response: Thank you. We add the definition of TIBF in the revised manuscript.

-In the independent variables paragraph, could you please add the categorization used for each variable in the analyses?

Author’s response: Thank you. We incorporate the categorization of each variables (see table 1 in the revised manuscript). 

-Why did you categorize maternal age in that way? Please, explain the rationale for those ranges. Consider recategorizing the variable into less categories since in the discussion section you describe just "older and younger mothers".

Author’s response: Dear reviewer thank you for raising this important issue. We did not re-categorize the age group for our analysis; we used the DHS categorization as it is. Many studies done on DHS data use this categorization. In addition, in the interpretation and in the discussion section we consider your comment. Dear reviewer we say older age because we compared with younger age groups (all consecutive age groups are older as compared with 15-19 age groups). If this does not convince you, we are open to consider your concern again and categorize to less categories. 

-Please, note that in the independent variables paragraph you said “four community level variables” but there were 5 mentioned (including residence).

Author’s response: Thank you. We amend it to read “Six community level variables” in the revised manuscript. 

-Regarding the community level factors, based on what did you set a 50% cut-off point?

Author’s response: Since the distribution of the community level variables were not normally distributed (to use the mean value), we used the national median value (50%) as a cut-off point.

-Please, note that in table 1 you also presented a “Community poverty level” that was not described in methods section.

Author’s response: Thank you very much. We consider your comment (we added community poverty level in the revised manuscript, which was missed in the method section). 

-In table 3, What does AOR mean? It was not mentioned in statistical analysis section. Besides, is the confidence interval 95%? Please add this information.

Author’s response: Thank you. We consider your comment in the data management and statistical analysis section of the revised manuscript. 

5. Results:

-The following information was given before: “total weighted sample of 101815 women was used for our analysis.”

Author’s response: Thank you for the comment. We avoided the redundancy in the revised manuscript. 

-Please check the following sentence “Most (40.46%) of the respondents had no formal education” note that most of the participants (59.5% ) had some formal education.

Author’s response: Thank you. We amended it per your recommendation. 

-Observe that the main purpose of the study is to investigate associated factor of TIBF. However, in this section you used the following title: “Determinants of timely initiation of breastfeeding…”

Author’s response: Thank you for the comment. We amended to read, “Factors associated with timely initiation of breastfeeding …” in the revised manuscript.

-Concerning “…those from a higher risk cluster had 1.37 times higher odds of TIBF as compared to those individuals who come from the lower risk cluster” Please, clarify what a higher risk refers to? In this case higher odds of TIBF would be beneficial/expected, right? It seems contradictory to talk about higher risk of having 1.37 times higher odds of TIBF. See if you can reformulate the sentence.

Author’s response: Thank you for the comment. We consider your comment and amend accordingly in the revised manuscript (see line 172-175).

-The information given from line 172 to 177 is repeating what was described from line 178 on.

Author’s response: Thank you. We consider your comment and remove redundancy in the revised manuscript.

-In table 3, I think that null model column is not necessary.

Author’s response: Dear reviewer thank you for the comment. It is the case in the multilevel model and if we remove the null model in the table, the table will be incomplete and may mislead the readers. Therefore, we prefer to put it in Table 3. We are open to remove the column showing the null model if you are not still convinced. 

-Please, review the labels of the categories in the wealth index. You can name the quintiles from the first (lowest) to the fifth (highest), instead of “the lowest and the second quintile”. It will be useful for describing this factor in the results and discussion section.

Author’s response: Thank you. We consider your comment in the revised manuscript.

6. Discussion and conclusion: 

-Please, conclude the first paragraph of the discussion. You compared the results of the study with other literature, but what does 58% of TIBF mean? Does it comply with the recommendation? (WHO reference: Infant and Young Child Feeding. A tool for assessing national practices, policies and programmes. Geneva. 2003).

Author’s response: Thank you. We consider your comment in the revised manuscript 

-The text from line 214 to 219 has already been described in the results section.

Author’s response: Thank you. We consider your comment and avoid redundancy in the revised manuscript. 

-In relation to “This is because educated mothers might have exposure to appropriate breastfeeding practices” could you explain it better?

Author’s response: Thank you. We explained it in a better way in the revised manuscript. 

In relation to “Contrary to our expectation, mothers who have been exposed to media had lower odds of TIBF as compared to their counterparts.” Why “contrary to our expectation”? What about the aggressive advertising of infant formula industry, milk substitutes, teats, and bottles regarding the media?

Author’s response: Really thank you for your concern and giving direction. We amended this statement accordingly in the revised manuscript.

-The discussion on community levels of ANC utilization and health facility delivery is not clear. Can you give more details and expand the discussion?

Author’s response: Thank you. We discussed these community level variables in detail in the revised manuscript.

-Line 292, determinants of TIBF, or associated factors?

Author’s response: We amended it to read, “Factors associated with TIBF” 

-It is important that the discussion includes not only the comparison of the results with other studies and the authors conclusions, but also the theoretical basis for these conclusions. Please, support all your conclusions with bibliographic references.

Author’s response: Thank you. We consider your comment and put references in the revised manuscript accordingly. 

-Please, review the conclusion: what was found in the study and the recommended "special emphasis", repeats the information.

Author’s response: Thank you again for raising this issue. We revised the conclusion section in the revised manuscript. 

7. References and general comments

-Please, check all the references and make sure the institutions names are consistently cited. (Organization WH., WHO., World Health Organization, Unicef., UNICEF.?)

Author’s response: We revise the reference section in the revised manuscript. 

-Please, remove the space between lines (for example, between line 56 and 57). Check all the text and make sure the format is consistent.

Author’s response: Thank you. We check the overall manuscript and we amended any errors encountered. 

-Please, avoid the following text structure (mistimed/unintended, inappropriate/suboptimal, physicians/midwives). If possible, choose one word or use “and” when necessary.

Author’s response: Thank you. We consider your comment in the revised manuscript. 

-Overall: I recommend editing the text to achieve a more appropriate grammar structure and scientific language.

Author’s response: Thank you. We extensively read our manuscript and edit it for language usage, spelling, and grammar after consulting our colleagues and language experts in our university.

Reviewer #2:

1. Abstract

Line 19 …’despite its major implication…’ perhaps it should read ‘despite THE major implications of…’ – However, I would suggest having this first line revised in its totality.

Author’s response: Thank you. We consider your comment in the revised manuscript.

2. Methodology

The authors have stated that they selected DHS data from 35 countries. Malawi, for example, has the most recent DHS data in 2015-16, why did the authors choose to analyse the 2011 (which I suppose should be 2010 data) instead of the most recent data?

Author’s response: Dear reviewer thank you for raising this important concern. We were used the 2015/16 survey data for Malawi, however we reported as we used the 2011 survey data. Therefore, in the revised manuscript and figure, we checked and modify to read, “2015/16”. 

The analysis is a pooled one, with 35 different countries having their respective weighting variables according to their sample selection. It would be appropriate for the authors to describe whether they generated pooled weights and briefly explain how this was done.

Author’s response: Dear thank you for raising this important issue. We appended the data sets and we apply weighting. That is weighting was conducted using the primary sampling unit variable, stratification variable, and the weight variable after appending the DHS data sets.

Did the authors check for multicollinearity of their models?

Author’s response: Thank you very much. Variance inflation factor (VIF) was used to assess Multicollinearity and there was no Multicollinearity between explanatory variables, with a mean VIF of 1.81 (the minimum and the maximum VIF was 1.01 and 3.96 respectively) (see line 113-115).

Under the data analysis section, authors aught to mention that adjusted odds ratio and 95% CI were reported…

Author’s response: Thank you. We mentioned it in data management and analysis section of the revised manuscript. 

3. Results

This is just a 'cosmetic' comment in the presentation of the results: It would look great if the fixed effects are presented first then random effects later thus Table 2 could be Table 3 while the current Table 3 may be changed to Table 2.

Author’s response: Thank you. We accepted your comment in the revised manuscript. 

4. Discussion

Line 285-290 – More explanation needed and if possible, provide some refs. Why would women coming from communities with high ANC utilization more likely to develop a norm of early BF initiation?

Author’s response: Thank you. We consider your comment in the revised paper. Moreover, women coming from communities with high ANC utilization are more likely to develop a norm of early BF initiation since there is higher probability of sharing information regarding infant feeding practices among mothers in the community. 

5. In the conclusion section, the authors have mention that the prevalence of TIBF was relatively lower, ,, lower compared to what? What are they comparing this prevalence to?

Author’s response: Thank you. We said low based on the clinical impact (since it is a major public health problem) and comparison with the other previous studies. However, we consider your comment and amend to read, “The prevalence of TIBF in SSA was good, consistent with the WHO rating of the TIBF practice as good”

---

## [Decision Letter · Decision Letter 1]

21 Jan 2021

PONE-D-20-27660R1

Timely initiation of breastfeeding and associated factors among mothers having children less than two years of age in sub-Saharan Africa: a multilevel analysis using recent Demographic and Health Surveys Data

PLOS ONE

Dear Dr. Teshale,

Thank you for submitting your manuscript to PLOS ONE. After careful consideration, we feel that it has merit but does not fully meet PLOS ONE’s publication criteria as it currently stands. Therefore, we invite you to submit a revised version of the manuscript that addresses the points raised during the review process.

In addition to the comments below, I highlight that:

As nicely pointed out by Reviewer #2, it is important to consider performing a three-level analysis to assess between-country heterogeneity, in view of the large interval of TIBF among SSA countries.While TIBF rate in SSA countries was "good" overall, according to WHO all babies (except in few cases where BF is contraindicated) should benefit from breastfeeding early in life. As indicated by Reviewer #1, such a finding should consider the great variation in SSA [note here that results from the three-level analysis could be quite informative] and also the great room for improvements in breastfeeding, including actions for the protection, promotion and support of this practice. Please revise the manuscript throughout.After revising your analysis, please make sure to properly reference the hypothesis drawn and/or underlying mechanisms suggested for the observed associations while discussing your findings.An extensive review of the abstract is needed (in up to 300 words), as well as of the structure of the discussion section.

We look forward to receiving your revised manuscript.

Kind regards,

Bárbara Hatzlhoffer Lourenço, Ph.D.

Academic Editor

PLOS ONE

Reviewers' comments:

Reviewer's Responses to Questions

**Comments to the Author**

1. If the authors have adequately addressed your comments raised in a previous round of review and you feel that this manuscript is now acceptable for publication, you may indicate that here to bypass the “Comments to the Author” section, enter your conflict of interest statement in the “Confidential to Editor” section, and submit your "Accept" recommendation.

Reviewer #1: (No Response)

Reviewer #2: All comments have been addressed

2. Is the manuscript technically sound, and do the data support the conclusions?

Reviewer #1: Yes

Reviewer #2: Yes

3. Has the statistical analysis been performed appropriately and rigorously? 

Reviewer #1: Yes

Reviewer #2: Yes

4. Have the authors made all data underlying the findings in their manuscript fully available?

Reviewer #1: Yes

Reviewer #2: Yes

5. Is the manuscript presented in an intelligible fashion and written in standard English?

Reviewer #1: Yes

Reviewer #2: Yes

6. Review Comments to the Author

Reviewer #1: The authors followed most of the reviewer's suggestions; however, some points still need to be reviewed, especially in the discussion section. The general comments below may guide the authors:

Please, check with the journal editor the number of words allowed in the abstract. In "Submission guidelines" - https://journals.plos.org/plosone/s/submission-guidelines#loc-abstract - Parts of submission - abstract, says: Not exceed 300 words.

Introduction:

Please, could you review the following sentence? “Even though there is a study conducted on the prevalence of TIBF in SSA [14], without identifying the hindering factors of TIBF, it is unlikely that TIBF can occur and reduce under-five mortality.” Could you briefly explain this better? I think the sentence is not clear.

Discussion:

-Please, review all the paragraphs and support your conclusions with bibliographic references. See an example:

“In this study, EBF was higher among multiparous mothers…. Cultural practices of early introduction of teas, water, and foods along with BM, probably have a greater impact on practices with the first child when mothers are not well supported for exclusive breastfeeding (reference). Thus, mothers with previous experience in the breastfeeding process and possibly older, are usually more mature in what concerns care and feeding of a child when compared to primiparous women.”

-I thank the authors for the English revision of the text. However, in the discussion section, I think the text could be more fluid. Please note that all paragraphs discussing the associated factors have the same structure: "This is because ..., This may be because ..., This might be because ..., This might be due to ..., This could be because ... ".

-I recommend the authors to further discuss the following factors: type of birth (single/multiple) and place of residence.

-Lastly, TIBF rate was good according to WHO (countries are consider good when at least 50% of the babies initiate BF in the first hour of life), but it is important to note that also according to WHO all babies (except when BF is contraindicated - few cases) should benefit from breastfeeding early in life. So, although the TIBF rate found in the study was good, it could be improved, especially because you found huge variation between countries. I think some words about it will complete your idea when you talk about "to plan appropriate strategies and implement interventions." Please, see if this makes sense to you.

Thank you for your previous response letter and for accepting most of the suggestions!

Reviewer #2: Thank you, authors, for adequately addressing my previous comments. I am following up with a few more comments that may need your consideration.

Abstract

Line21-22, the sentence should be rewritten as it stands, it may mean that the SDG on reduction of <5 mortality solely depends on optimal breastfeeding. In as much as optimal breastfeeding is an important component in achieving this SDG goal, there are other indicators such as vaccination that are also important. Maybe authors can just state that ‘optimal breastfeeding is one of the key components that may help achieve reduction of <5 mortality….’

In the abstract conclusion (Line 42), authors state that the prevalence of TIBF was GOOD in SSA. ‘Good’ based on what standard?

Introduction

Line 70-72 in the introduction section is not clear.

Methods

- Please state that the most recent DHS data was selected for analysis from each country specifically for those countries that have more than 1 surveys.

- Line 81-82; ‘after appending….’ I think this is misplaced. Perhaps include it in the data management section.

- I want clarification whether authors constructed new cluster and stratum variables and whether the weights were redefined considering that they had appended the datasets from several countries.

Results

For the pooled estimates, did the authors adjust for survey (i.e., country)? I think the results would be more interesting if a variable called ‘country/survey’ is included.

Follow up suggestion on the above comment: The authors did a two-level analysis, where individuals, and communities were analysed. As this survey pooled several countries, a three-level analysis would have been more interesting to not only observe community-level heterogeneity, but also assess between country heterogeneity

7. PLOS authors have the option to publish the peer review history of their article (what does this mean?). If published, this will include your full peer review and any attached files.

Reviewer #1: No

Reviewer #2: No

---

## [Author Response · Author response to Decision Letter 1]

29 Jan 2021

Date: January 29, 2021 

Response to editor and reviewer

Title: Timely initiation of breastfeeding and associated factors among mothers having children less than two years of age in sub-Saharan Africa: a multilevel analysis using recent Demographic and Health Surveys Data

Manuscript number: PONE-D-20-27660R1

Dear editor and reviewers thank you for raising the important concerns for the betterment of the manuscript. Here is the point-by-point response from the authors. 

Response to editor 

1. As nicely pointed out by Reviewer #2, it is important to consider performing a three-level analysis to assess between-country heterogeneity, in view of the large interval of TIBF among SSA countries.

Author’s response: Thank you for the important comment you raised and your recommendation. We have tried to did a three level analysis to encounter the between country heterogeneity and unfortunately the output is not generated (we have convergence problem). We hope that as far as we account community-level heterogeneity the estimates are not that much different. Dear editor, we have acknowledged this as the limitation in the revised manuscript. 

2. While TIBF rate in SSA countries was "good" overall, according to WHO all babies (except in few cases where BF is contraindicated) should benefit from breastfeeding early in life. As indicated by Reviewer #1, such a finding should consider the great variation in SSA [note here that results from the three-level analysis could be quite informative] and also the great room for improvements in breastfeeding, including actions for the protection, promotion and support of this practice. Please revise the manuscript throughout.

Author’s response: Thank you. We have considered your concern in the revised manuscript. Considering the variations of TIBF in SSA and the WHO recommendation of all babies should benefit from breastfeeding early in life, we amend it to read, “TIBF in SSA countries was low”. 

3. After revising your analysis, please make sure to properly reference the hypothesis drawn and/or underlying mechanisms suggested for the observed associations while discussing your findings.

Author’s response: Thank you. We have revised the underlying mechanisms suggested for the observed associations while discussing our findings. 

4. An extensive review of the abstract is needed (in up to 300 words), as well as of the structure of the discussion section.

Author’s response: Thank you. We have considered your comment in the revised manuscript. We minimized the abstract to 300 words and we review and amend the structure of the discussion section. 

Response to reviewer 

Reviewer #1:

1. Please, check with the journal editor the number of words allowed in the abstract. In "Submission guidelines" - https://journals.plos.org/plosone/s/submission-guidelines#loc-abstract - Parts of submission - abstract, says: Not exceed 300 words.

Author’s response: Thank you. We amend the abstract section (we minimize the number of words to 300).

2. Introduction:

Please, could you review the following sentence? “Even though there is a study conducted on the prevalence of TIBF in SSA [14], without identifying the hindering factors of TIBF, it is unlikely that TIBF can occur and reduce under-five mortality.” Could you briefly explain this better? I think the sentence is not clear.

Author’s response: Thank you. We make it clear in the revised manuscript.

3. Discussion:

-Please, review all the paragraphs and support your conclusions with bibliographic references. See an example: “In this study, EBF was higher among multiparous mothers…. Cultural practices of early introduction of teas, water, and foods along with BM, probably have a greater impact on practices with the first child when mothers are not well supported for exclusive breastfeeding (reference). Thus, mothers with previous experience in the breastfeeding process and possibly older, are usually more mature in what concerns care and feeding of a child when compared to primiparous women.”

Author’s response: Thank you. We have reviewed all the paragraphs and we have putted the conclusions we made with bibliographic references. 

-I thank the authors for the English revision of the text. However, in the discussion section, I think the text could be more fluid. Please note that all paragraphs discussing the associated factors have the same structure: "This is because ..., This may be because ..., This might be because ..., This might be due to ..., This could be because ... ".

Author’s response: Really thank you for the important comment you raised. We consider your issue in the revised manuscript.

-I recommend the authors to further discuss the following factors: type of birth (single/multiple) and place of residence.

Author’s response: Thank you. The above stated variables are discussed in the revised manuscript. 

-Lastly, TIBF rate was good according to WHO (countries are consider good when at least 50% of the babies initiate BF in the first hour of life), but it is important to note that also according to WHO all babies (except when BF is contraindicated - few cases) should benefit from breastfeeding early in life. So, although the TIBF rate found in the study was good, it could be improved, especially because you found huge variation between countries. I think some words about it will complete your idea when you talk about "to plan appropriate strategies and implement interventions." Please, see if this makes sense to you.

Author’s response: Thank you for your comment and direction. We have considered your recommendation and revised the statement accordingly. 

Reviewer #2: 

1. Abstract

Line21-22, the sentence should be rewritten as it stands, it may mean that the SDG on reduction of <5 mortality solely depends on optimal breastfeeding. In as much as optimal breastfeeding is an important component in achieving this SDG goal, there are other indicators such as vaccination that are also important. Maybe authors can just state that ‘optimal breastfeeding is one of the key components that may help achieve reduction of <5 mortality….’

Author’s response: Dear reviewer, thank you for the nice comment you raised. We have considered your comment as well as recommendation and amended the statement accordingly. 

In the abstract conclusion (Line 42), authors state that the prevalence of TIBF was GOOD in SSA. ‘Good’ based on what standard?

Author’s response: Thank you. It is amended to read “The prevalence of TIBF in SSA was low” and we said low based on the huge variations between countries and according to the WHO recommendation that all babies should benefit from breastfeeding early in life. 

2. Introduction

Line 70-72 in the introduction section is not clear.

Author’s response: Thank you. We make it clear in the revised manuscript (see line 67 and 68).

3. Methods

- Please state that the most recent DHS data was selected for analysis from each country specifically for those countries that have more than 1 surveys.

Author’s response: Thank you. We have considered it in the revised manuscript (see line 76 & 77).

- Line 81-82; ‘after appending….’ I think this is misplaced. Perhaps include it in the data management section.

Author’s response: Thank you. We considered your comment in the revised manuscript (The first sentence of the data management and analysis section, line 99). 

- I want clarification whether authors constructed new cluster and stratum variables and whether the weights were redefined considering that they had appended the datasets from several countries.

Author’s response: Dear reviewer, thank you for the important concern you raised. As you know, the weighting variables are coded in similar in each DHS we have weighted the data using sample weight, v005 that is helpful to adjust over and under sampling and the SVY set command to account for the complex survey design and generalizability of the findings.

4. Results

For the pooled estimates, did the authors adjust for survey (i.e., country)? I think the results would be more interesting if a variable called ‘country/survey’ is included.

Dear reviewer: We have weighted the data using sample weight, v005 ,that is helpful to adjust over and under sampling and the SVY set command to account for the complex survey design and generalizability of the findings.

Follow up suggestion on the above comment: The authors did a two-level analysis, where individuals, and communities were analysed. As this survey pooled several countries, a three-level analysis would have been more interesting to not only observe community-level heterogeneity, but also assess between country heterogeneity

Author’s response: Dear reviewer we have weighted the data to adjust sampling and to account for complex survey design. However, we are unable to conduct a three level analysis since we have convergence difficulty (even we tried to conduct a three level analysis to identify both community-level heterogeneity and country heterogeneity but we are unable to get the output). Dear reviewer, many multilevel studies conducted using pooling of the DHS data also adjust for community level heterogeneity only due to similar issue. Therefore, we acknowledge it in the revised manuscript as limitation of our study (see line 273 &274). Hope this convinces you, however, we are open to accept directions and recommendations if you have.

---

## [Decision Letter · Decision Letter 2]

9 Mar 2021

Timely initiation of breastfeeding and associated factors among mothers having children less than two years of age in sub-Saharan Africa: a multilevel analysis using recent Demographic and Health Surveys Data

PONE-D-20-27660R2

Dear Dr. Teshale,

We’re pleased to inform you that your manuscript has been judged scientifically suitable for publication and will be formally accepted for publication once it meets all outstanding technical requirements.

Kind regards,

Bárbara Hatzlhoffer Lourenço, Ph.D.

Academic Editor

PLOS ONE

Additional Editor Comments (optional):

Prior to publication, reviewers have suggested that an English review would benefit the readability of the manuscript and, more specifically:

1. In the discussion section, conclusions for "unintended pregnancies and average size of the child at birth" should be supported with appropriate literature;

2. In the abstract and conclusion, rate of TIBF was indicated as low, but in the discussion section it was indicated as good --it would be important to keep suc statements consistent throughout the text.

Reviewers' comments:

Reviewer's Responses to Questions

**Comments to the Author**

1. If the authors have adequately addressed your comments raised in a previous round of review and you feel that this manuscript is now acceptable for publication, you may indicate that here to bypass the “Comments to the Author” section, enter your conflict of interest statement in the “Confidential to Editor” section, and submit your "Accept" recommendation.

Reviewer #1: (No Response)

Reviewer #2: All comments have been addressed

2. Is the manuscript technically sound, and do the data support the conclusions?

Reviewer #1: Yes

Reviewer #2: Yes

3. Has the statistical analysis been performed appropriately and rigorously? 

Reviewer #1: Yes

Reviewer #2: No

4. Have the authors made all data underlying the findings in their manuscript fully available?

Reviewer #1: Yes

Reviewer #2: No

5. Is the manuscript presented in an intelligible fashion and written in standard English?

Reviewer #1: Yes

Reviewer #2: Yes

6. Review Comments to the Author

Reviewer #1: Dear authors,

Prior to publication, please, review the following suggestions:

-In the discussion section, the conclusions of "unintended pregnancies and average size of the child at birth" should also be supported with bibliographic references.

In the abstract and conclusion section you stated that the rate of TIBF was low, but in the discussion section you said it was good. Please, keep your conclusion consistent throughout the text.

Reviewer #2: Thank you authors for addressing my previous comments. The paper is technically sound. Minor English edits would be essential but otherwise, I am happy with the responses

7. PLOS authors have the option to publish the peer review history of their article (what does this mean?). If published, this will include your full peer review and any attached files.

Reviewer #1: No

Reviewer #2: No

---

## [Editor Report · Acceptance letter]

12 Mar 2021

PONE-D-20-27660R2 

Timely initiation of breastfeeding and associated factors among mothers having children less than two years of age in sub-Saharan Africa: a multilevel analysis using recent Demographic and Health Surveys Data 

Dear Dr. Teshale:

I'm pleased to inform you that your manuscript has been deemed suitable for publication in PLOS ONE. Congratulations! Your manuscript is now with our production department. 

Kind regards, 

on behalf of

Dr. Bárbara Hatzlhoffer Lourenço 

Academic Editor

PLOS ONE